# Are Bioactive Molecules from Seaweeds a Novel and Challenging Option for the Prevention of HPV Infection and Cervical Cancer Therapy?—A Review

**DOI:** 10.3390/ijms22020629

**Published:** 2021-01-10

**Authors:** Marius Alexandru Moga, Lorena Dima, Andreea Balan, Alexandru Blidaru, Oana Gabriela Dimienescu, Cezar Podasca, Sebastian Toma

**Affiliations:** 1Department of Medical and Surgical Specialties, Faculty of Medicine, Transylvania University of Brasov, 500019 Brasov, Romania; moga.og@gmail.com (M.A.M.); dimienescu.oana@gmail.com (O.G.D.); caesarpodaska@yahoo.com (C.P.); 2Department of Fundamental, Prophylactic and Clinical Sciences, Faculty of Medicine, University Transilvania of Brasov, 500019 Brasov, Romania; lorenadima@yahoo.com (L.D.); sebitom2002@yahoo.com (S.T.); 3Department of Surgical Oncology, Oncological Institute “Al. Trestioneanu” of Bucharest, University of Medicine and Pharmacy Carol Davila Bucharest, 020021 Bucharest, Romania

**Keywords:** HPV infection, anticancer, seaweeds, cervical cancer, carcinogenesis process

## Abstract

Cervical cancer represents one of the leading causes of cancer-related death in women all over the world. The infection with human papilloma virus (HPV) is one of the major risk factors for the development of premalignant lesions, which will progress to cervical cancer. Seaweeds are marine organisms with increased contents of bioactive compounds, which are described as potential anti-HPV and anti-cervical cancer agents. Our study aims to bring together all the results of the previous studies, conducted in order to highlight the potency of bioactive molecules from seaweeds, as anti-HPV and anti-cervical agents. This paper is a review of the English literature published between January 2010 and August 2020. We performed a systematic study in the Google Academic and PubMed databases using the key words “HPV infection”, “anticancer”, “seaweeds”, “cervical cancer” and “carcinogenesis process”, aiming to evaluate the effects of different bioactive molecules from marine algae on cervical cancer cell lines and on HPV-infected cells. Only original studies were considered for our research. None of the papers was excluded due to language usage or affiliation. Recent discoveries pointed out that sulfated polysaccharides, such as dextran sulfate heparan or cellulose sulfate, blocked the ability of HPV to infect cells, and inhibited the carcinogenesis process. Carrageenans inhibited the virions of HPV from binding the cellular wall. Fucoidan induced the growth inhibition of HeLa cervical cells in vitro. Heterofucans exhibited antiproliferative effects on cancer cell lines. Terpenoids from brown algae are also promising agents with anti-cervical cancer activity. Considering all the results of the previous studies, we observed that great amounts of bioactive molecules from seaweeds could treat both unapparent HPV infection and clinical visible disease. Furthermore, these molecules were very efficient in the treatment of invasive cervical carcinomas. In these conditions, we consider seaweeds extracts as a novel and challenging therapeutic strategy, and we hope that our study paves the way for further clinical trials in the field.

## 1. Introduction

In Europe, cervical cancer is the second most prevalent cause of death in young women, registering almost 24,400 deaths annually [1]. Human papillomavirus infection progresses to premalignant lesions, which finally result in cervical cancer. In the last few years, a wide range of studies on the natural history of HPV have highlighted the importance of the persistence of this virus in the development of invasive cervical cancer. Human papillomavirus belongs to the group of non-enveloped DNA (deoxyribonucleic acid) viruses, which are able to infect mucosal tissues and skin, in humans and other vertebrates [2]. This virus can exist in either the genital and urinary tract, throat or mouth [3]. The infection with HPV is usually asymptomatic and self-limiting. Inflammation represents a complex physiological response of the organism to harmful factors, such as infections, cellular changes or environmental changes [4]. In these conditions, there is a subset of persistently infected women whose lesions can progress to premalignant lesions or invasive cervical carcinomas [5]. There are several cofactors associated with cervical cancer development, such as prolonged oral contraceptives use, sexually transmitted infections, smoking, etc., but their role remains minor in comparison with HPV infection and persistence [6]. Approximately 43% of vulvar, 70% of vaginal and 100% of cervical tumors are associated with HPV infection [7]. Although HPV is usually transmitted by sexual intercourse, other non-sexual routes such as perinatal vertical transmission or physical contact have been described [8].

In the last decade, more and more researchers from all over the world have shown an increased interest in the prevention and treatment of HPV infection and cervical cancer. Seaweeds are natural organisms with increased contents of polysaccharides and other bioactive compounds, known for their multitude of health-promoting effects. Their potential as anticancer and antiviral agents attracted the attention of the investigators.

The interest of consumers regarding health food products has significantly increased in the recent period. Algae are marine organisms able to provide a wide range of bioactive compounds with high curative potential. These organisms have been traditionally used as a foodstuff in Asian countries for centuries. Nowadays, the biggest consumers of seaweeds are Japan, China, Korea, Hawaii, Indonesia, and Philippines. Epidemiological studies have shown a lower prevalence of diseases such as coronary heart disease and diet-related cancers in countries with high seaweed consumption [9,10]. However, there is no scientific evidence showing less incidence of cervical cancers or HPV infections in these populations. Recent discoveries have shown that seaweeds could represent some of the most important reservoirs of natural therapeutic agents for many pathologies [11]. Their diverse origin and universal availability, along with their natural abundance in the marine environment, make them a prolific source of biologically functional compounds [12].

Marine algae are classified into two groups: macroalgae, or seaweeds, and microalgae. Microalgae is a group that includes approximately 5000 different species, such as blue-green algae, bacillariophyte or dinoflagellates [12,13]. Seaweeds include more than 6000 species [13] that grow in deep seas, up to a depth of 180 m. Sometimes, they can be found in shallow coastal waters [14]. The algae that grow in saline waters or river mouths are very rich in sulfated polysaccharides and other biologically active compounds, which transform seaweeds into novel pharmaceutical agents for humans [15].

Seaweeds are classified based on their sizes, forms, and pigments. They contain natural pigments that divide them into the following categories: red algae (*Rhodophyceae*), green algae (*Chlorophyceae*), and brown algae (*Phaeophyceae*). Their colors are due to the presence of phycoerythrin, chlorophyll, and fucoxanthin. Humans cultivated several species of seaweeds, and this biotechnology is continuously developing. Recent advances in the cultivation of seaweeds include the establishment of cell and tissue cultures that can biologically synthesize the desired bioactive compounds, under strictly controlled conditions [16]. 

Marine macroalgae contain a wide range of bioactive compounds and exhibit many beneficial effects, such as antiviral, anticancer, anti-inflammatory, anticoagulant, antimicrobial, or hypocholesterolemic activities [17]. Figure 1 summarizes several of the health-promoting effects of seaweeds and their bioactive compounds which exert these activities.

This paper focuses on all the previous studies based on the anti-HPV effect of seaweeds, and on their efficiency against cervical cancer. Moreover, we aimed to find a common denominator of the mechanisms of action of these natural sources of bioactive molecules against cervical cancer, premalignant cervical lesions, and HPV infection.

## 2. Bioactive Compounds of Seaweeds

Seaweeds are known for their natural richness in minerals such as sulfur, iron, iodine, potassium, vitamins, and bioactive molecules such as polyunsaturated fatty acids, natural pigments, proteins, carbohydrates, lipids, and especially sulfated polysaccharides [18,19]. Several studies have revealed that seaweeds also contain a large amount of antioxidants, such as polyphenols [20]. The biochemical composition of seaweeds usually depends on the geographic area and seasonal conditions [21]. The variations in the concentration of these compounds are influenced by different environmental factors such as light, carbon dioxide (CO_2_), pH, contaminants, and temperature [22]. 

In marine organisms, polysaccharides are widely distributed. The content of polysaccharides in seaweeds is almost 50% of the dry weight [23]. In green seaweeds the principal polysaccharides are ulvans, red seaweeds abound in agarans and carrageenans, while in brown seaweeds fucans, laminarin, and alginates prevail [24].

Alginates are acidic polysaccharides composed of poly-D-mannuronic acid and residues of D-mannuronic acid and D-glucuronic acid, with a central backbone of poly-D-glucuronic acid [25]. Fucans, along with alginates, can be found in the cell walls of brown seaweeds. These sulfated polysaccharides entail ramifications for each two or three of the residues of fucose, and also possess a central core of sulfated fucose [26]. 

Agarans are polysaccharides extracted from red seaweeds, which contain 4-linked α-galactose residues of the L-series. Carrageenans can be found in red seaweeds and their composition consists of 4-linked α-galactose residues of the D-series [27]. These molecules possess a broad spectrum of antiviral, anticoagulant, anticancer activities, and various studies have referred their anti-HPV effects [23,28].

Seaweeds are also a rich source of polyphenols, such as catechins, flavonols, and phlorotannins in particular. In green and red algae, flavonoids, phenolic acids and bromphenols are the largest proportion of phenolic compounds. Phlorotannins represent a group of complex polymers of phloroglucinol and can be found only in brown algae [20]. *Sargassum thunbergii, Ecklonia cava*, *Ecklonia stolonifera*, *Undaria pinnatifida*, *Ecklonia kurome*, and *Hizikia fusiformis* are several of the most rich algae in phlorotannins [29]. Phlorotannins exist in soluble or cell wall-bound forms, and are necessary for the physiological integrity of the algae. These phytochemicals have attracted much attention due to their potential health benefits in numerous human diseases, such as anticancer, antiviral, antimicrobial, and anti-inflammatory activities. However, the antioxidant activity of seaweed phlorotannins has gained the most interest from the researchers [20]. 

There is a huge diversity in high-value natural edible pigments from seaweeds. Of these, xanthophylls (antheraxanthin, fucoxanthin, lutein, neoxanthin, violaxanthin, zeaxanthin), carotenoids and chlorophylls are the most abundant classes of pigments [30]. Scientific evidence has shown that marine algae pigments are able to exhibit several bioactive properties, such as antioxidant, immune modulatory, antiangiogenic, anti-inflammatory, etc. Chlorophyll is a ubiquitous pigment in green algae and it is non-covalently attached to proteins to form pigment–protein complexes. It is a tetrapyrrole formed by four porphyrin rings, with a central magnesium ion [31]. Phycobiliproteins are a family of water-soluble pigments, mainly found in *Rhodophyceae* and *Cyanophyceae* families. Their chemical composition is almost similar to the composition of chlorophyll, but it occurs as an open-chain similar to phytochrome, and lacks the central magnesium ion [32]. Due to their high solubility and hydrophilicity, phycobiliproteins find numerous applications in molecular biology and diagnostics [31]. Carotenoids are a class of naturally occurring yellow, orange and green pigments widely distributed in edible algae. The structure of carotenoids consists of eight isoprene units of repeating isoprenoid, resulting in a 40-carbon molecule that contains conjugated double bonds [33]. When consumed by humans and animals, these pigments exert strong antioxidant activities that can offer protection against various diseases, including cancer [31]. 

The total protein, lipid, and individual fatty acid contents of edible seaweeds are still debated. According to Sanchez-Machado et al. [34], the total lipid content of various marine algae range between 0.70 ± 0.09 and 1.80 ± 0.14 g/100 g dry weight. C20:4ω6, C16:0, C20:5ω3 and C18:1ω9 were found to be the most abundant fatty acids. Despite the fact that polyunsaturated fatty acids predominate in brown seaweeds, and saturated fatty acids predominate in red seaweeds, both groups are rich sources of ω3 and ω6 acids. The protein content varies from 5.46 ± 0.16 to 24.11 ± 1.03 g/100 g dry weight in different species of marine algae. 

In Table 1 we summarize the principal bioactive compounds found in various types of seaweeds. 

## 3. Anti-HPV and Anticancer Mechanisms of Seaweeds Compounds

### 3.1. Anti-HPV Mechanisms of Seaweeds Polysaccharides

Nowadays, marine plant-derived polysaccharides have become a precious resource of potential anti-HPV drugs due to their ability to inhibit the virus in different phases of the infection process. Furthermore, several seaweed polysaccharides have been described as immunomodulators [48]. 

Carrageenan is a sulfated polysaccharide extracted from red algae, which has demonstrated its anti-HPV efficiency in vitro. Its mechanisms of action mainly consist of the inhibition of virus adsorption and internalization into the host cells [49]. Carrageenan inhibited HPV withy nearly a thousand-fold higher potency than heparin, which is a highly effective anti-HPV agent [50]. λ- cyclized μ/ι-carrageenans inhibited the attachment, while λ-, ι-carrageenans inhibited the internalization, uncoating, transcription and replication of HPV [48]. Carrageenan usually acts by inhibiting the binding of HPV virions to the cellular wall. Then, it blocks the subsequent infection through a post-attachment heparin sulfate-independent effect [11]. 

The attachment of HPV virions to cultured cell lines is mediated by various interactions between the virion and heparan sulfate, which is a glycosaminoglycan found on the cell surface. Carrageenan and other sulfated polysaccharides block the viral infection process by chemically mimicking the heparan sulfate. Then, they compete with healthy cells against initial virion attachment [51]. Based on these findings, carrageenan may be an innovative therapeutic tool against HPV infection, and carrageenan-based sexual lubricant gels could block the sexual transmission of HPV [52]. 

Fucoidan is a marine heparinoid polysaccharide extracted from brown algae that efficiently inhibited HPV infection in vitro, having a IC50 value of 1.1 µg/mL. Despite being not as efficient as carrageenan, fucoidan also deserves further investigation as a novel therapeutic agent in the future for cervical cancer and HPV infection [50]. 

Heparin is a glycosaminoglycan which consists of a variably sulfated repeating disaccharide unit, and can be extracted from marine algae. Various strains of HPV have been reported to be able to bind heparin sulfate, as a low affinity co-receptor. High-risk HPV16 is able to bind heparan sulfate from the cells surface through viral capsid protein L1, and it cannot infect the cervical cells without this molecule [53]. Heparin sulfate can also mimic the receptors for HPV. Bienkowska et al. [54] have pointed out that the binding of L1 protein to heparan sulfate makes the cyclophilin B and L2 viral protein, to promote the cyclophilin B-mediated conformational change of the L2 protein. Secondary to this event, HPV can invade the host cells. It has been proven that pseudoinfection was inhibited by heparin [55]. 

In conclusion, heparin is able to interfere with the initial attachment of the virions to the host cells, suggesting that these molecules could be very effective as anti-HPV drugs. Figure 2 illustrates the mechanisms through which seaweeds polysaccharides can inhibit the HPV infection process and can induce apoptosis in DNA-damaged cells with high potential for neoplastic transformation.

### 3.2. Anti-Cancer Mechanisms of Seaweeds Compounds

The anticancer effects of seaweeds compounds are exerted by various pathways, but a consensus in this field has still not been reached, and future studies are required in order to better describe these mechanisms. Based on the previous results, edible seaweeds exert protective effects against mammary, skin, cervical and intestinal cancer [56]. Terpenes, polysaccharides, and polyphenols are considered to be of major interest for their anti-cancer properties [57]. 

As a nutrient source, seaweeds can reduce the development of tumors due to their antioxidant properties. Considering that these organisms are one of the richest repositories of natural antioxidants among the marine resources, they are a feasible tool to block the progression of premalignant lesions to invasive cancer. The major anti-cancer pathways include antioxidation, tumoral cells apoptosis, and immune stimulation. Tumors generate significant quantities of free radicals that are usually accompanied by a lack of DNA repair mechanisms [58]. Reactive oxygen species (ROS) represent the most important factor for DNA damage and also induces high levels of oxidative stress [59]. For this reason, antioxidant agents, such as seaweeds, can inhibit the progression of abnormal cells to cancer. Chlorophyll, fucoxanthin and phycoerythrobilin from edible algae showed the strongest antioxidant activity [11].

The immunomodulatory activity of seaweed compounds was assessed in Swiss mice by analyzing the effects of sulfated polysaccharides isolated from *C. feldmannii*. The investigators observed that sulfated polysaccharides induced an increased production of both specific antibodies and OVA-specific antibodies. Moreover, a discreet hyperplasia of the lymphoid follicles of the white pulp in the spleen was reported, and was associated with the anti-cancer activity of seaweed sulfated polysaccharides [60]. Fucoidan was associated with a significant increase in the activity of natural killer (NK) lymphocytes. Moreover, it has been proven that this molecule is able to increase the production of Interferon gamma (IFN γ) by T cells and to modulate NK and T helper 1 (Th1) cell responses in leukemia mice [61]. 

In later stages of carcinogenesis, high doses of seaweeds extracts exert strong cytotoxic properties [62]. Elatol is a compound of red algae that induces cytotoxicity in tumor cell lines through cell cycle arrest and secondary apoptosis. Protein p53 and the caspase-cascade signaling system usually mediate the promotion of apoptosis. The apoptosis process consists of three events, activation, execution and cell deletion, and all of these stages are linked by caspases [63]. The prevention of cancer depends on p53. This protein triggers cell growth arrest and apoptosis, and also controls the proliferation of cells with damaged DNA that are susceptible to malignant transformation [64]. Seaweeds represent sources of many phytochemicals which trigger the apoptosis process. Phenolic compounds are composed from a single aromatic ring and possess a wide range of biological activities, including anti-cancer effects. Polyphenols from red edible seaweeds exert anti-cancer activity by inducing apoptosis, decreasing the oxidative stress and downregulating the endogenous estrogen biosynthesis [65]. Pheophorbide a is a product of chlorophyll, which could be used in the photodynamic therapy of cancer. It has been proven that Pheophorbide a is able to induce G0/G1 arrest in tumoral cells, leading to late apoptosis and DNA degradation under dark conditions [62]. This compound has not proven its utility in the treatment of cervical cancer, but according to the most recent scientific evidence, it is a potential glioblastoma-specific anti-cancer agent [62]. 

Marine algae polysaccharides are also known to cause cell death by inducing apoptosis. Protein caspases, Bax and Bcl-2 are frequently involved in this process. For example, Fucoidan was found to upregulate the pro-apototic proteins Bax, caspase 3 and caspase 9 in tumoral cells [66]. Porphyran induced apoptosis by increasing the activity of caspase-3 in colon cancer cell lines [67]. Additionally, this polysaccharide blocked cell proliferation in HeLa cells. The cell cycle was stopped in the G2/M phase via the expression of CDK1, p21, p53, and cyclin B1 [68]. 

Regarding the cytotoxic effects of carrageenan, a recent study has shown that κ-carrageenan and λ-carrageenan inhibited HeLa cells in vitro. These molecules arrested the cell cycle at specific phases, as follows: κ-carrageenan delayed the cell cycle in the G2/M phase and λ-carrageenan delayed the G1 and G2/M phases [69]. Moreover, κ-carrageenan combined with selenium, known as κ-selenocarrageenan, blocked the human hepatoma cell cycle in the S phase [70]. Degraded ι-carrageenan downregulated the Wnt/β-catenin signaling pathway and suppressed the tumor growth. Additionally, it induced apoptosis and arrested the G1 phase in human osteosarcoma cells [71]. 

Tumors need a high supply of oxygen and nutrients which can only be achieved by increased angiogenesis in the growing mass. Thus, various agents that target the angiogenesis process could block the ability of the tumors to grow [72]. In the literature, there is much evidence suggesting that the sulfated polysaccharides extracted from seaweeds, such as fucoidan and laminarin, could have anti-angiogenic effects. Angiogenesis is a three-dimensional process, and in vivo models such as mice or chick eggs were used to analyze the effects of sulfated polysaccharides against it [73]. A study conducted by Koyanagi et al. [74] pointed out the anti-angiogenic effect of fucoidan in a mouse model. Moreover, fucoidan extracted from *Undaria pinnatifida* was reported to reduce both vascular endothelial growth factor (VEGF) and other pro-angiogenic citokines, leading to a significant reduction in the tumor size of 4T1 tumors [75,76]. Plasminogen activator inhibitor-1 (PAI-1) is a protein secreted by the endothelial cells, and high levels are found in cancer patients. Pieces of evidence have shown that this molecule could be a good target for anti-cancer drugs [77]. It has been reported that the exposure to fucoidans, especially to those with higher sulphation levels, was correlated with decreased levels of PAI-1 secretion [78]. 

Hoffman et al. investigated the effects of sulfated laminarin on angiogenesis, using a chick chorioallantoic membrane assay. They observed that high doses of laminarin inhibited the angiogenesis process, but induced toxicity, leading to chick embryo deaths. A synergistic anti-tumor effect of this polysaccharide was described when it was combined with a cytotoxic agent [79].

## 4. Material and Method

This paper is a review of the English literature published in the last decade. We performed a systematic search in PubMed and Google Academic databases, using the key words “seaweeds”, “HPV infection”, and “cervical cancer” in order to evaluate the effects of various bioactive compounds extracted from edible seaweeds on cervical cancer cell lines and on HPV-infected cells. Only original studies were included in our research. None of the articles were excluded due to language usage or affiliation. After a systematic review, we identified 18 articles that fitted our area of interest and that will be further presented and discussed. 

## 5. Results and Discussion

### 5.1. Studies Regarding the Effects of Seaweeds Bioactive Compounds on HPV Infection

The vital role of HPV in the development of cervical cancer was previously discussed. After a systematic search, we found only three studies that pointed out the beneficial effects of seaweed bioactive compounds against HPV infection. According to a recent study, carrageenan from marine sources generated specific anti-cancer activities in female mice, after being vaccinated with HPV16 E7 peptide [80]. Additionally, this polysaccharide induced antigen-specific immune responses in the infected mice. Rodriguez et al. [81] used a gel containing 3% carrageenan in order to show if this polysaccharide is able to prevent HPV infection in vivo and in vitro. Carrageenan formulations were applied 2 h before the contact with the virus. The investigators observed that carrageenan provided durable protection from HPV infection in mouse models. Furthermore, they reported that seminal plasma did not interfere with the antiviral activity of carrageenan against HPV. 

Polymannuroguluronate sulfate (PMGS) is a polysaccharide extracted from brown seaweeds. According to Wang et al. [82], PMGS effectively blocked infection with HPV 16 and HPV 45 in BALB/c Nude Mice by targeting the L1 protein from the viral capsid. Additionally, PMGS penetrated HeLa cells and decreased the expression of viral oncogene proteins E6 and E7 of HPV.

In the literature, there is a lack of studies regarding the effects of seaweeds bioactive molecules on premalignant dysplastic lesions. We found only one study conducted by Santos et al. [83], which reported the effects of *Porphyra umbilicalis* on HPV-transgenic mice with pre-malignant dysplastic skin lesions. This seaweed was incorporated into the diet of HPV 16-infected mice, which have developed pre-malignant and malignant lesions, in order to determine whether this product can block the development of these lesions. After the follow-up period, the investigators reported that *Porphyra umbilicalis* blocked the development of pre-malignant lesions and exerted antigenotoxic activity against HPV-induced DNA damage [83].

In conclusion, despite this scientific evidence of the efficiency of seaweeds in the fight against HPV infection, we consider that further studies are needed to establish these natural agents as a functional food and to clarify the mechanisms whereby multistep carcinogenesis induced by HPV could be blocked. Another topic of high interest is represented, in our opinion, by the doses and the time of administration needed in order to obtain a protective effect. Moreover, the association of seaweeds, as dietary sources, with anti-HPV vaccine could represent a supplementary measure in the prevention of HPV infection, and future research on this topic could be of major interest for the biomedical community in the field of cervical cancer and HPV infection. 

### 5.2. Studies Regarding the Effects of Seaweeds Bioactive Compounds on Cervical Cancer

We found fifteen studies in the literature that fitted our inclusion criteria, which will be further presented and discussed. Unfortunately, all of these studies were conducted in vitro, using cancer cell lines and high doses of seaweed extracts. The only exception is represented by a study conducted in the year 2013 [84], which aimed to investigate whether the dietary intake of carotenoids, which are micronutrients from seaweeds, can influence the regression of low-grade cervical abnormalities. The prospective cohort included 391 patients with cervical intraepithelial neoplasia (CIN) grade 1–2 lesions, whose dietary intake included various doses of beta-carotene. The results indicated that of the 391 women, the regression, persistence and progression of the cervical premalignant lesions occurred in 218, 135 and 38, respectively. Moreover, it has been proven that maintaining a medium serum level of beta-carotene has a protective effect for the progression of cervical dysplasia. The main limitations of this study were the following: the dietary sources of carotenoids were not only seaweeds, and also the food intake contained not only carotenoids, but also other mixtures and nutrients. Another limitation was represented by the fact that the included subjects already had persistent HPV infection and the preventive role of carotenoids could not be assessed. 

Dewi and coworkers [85] used diluted extracts of *Gracilaria verrucosa,* aiming to demonstrate the anti-cancer effects of this seaweed on HeLa cell lines. They observed that hexane, ethanol, chloroform, and ethyl acetate extracts of *Gracilaria verrucosa* displayed significant cytotoxicity against HeLa cells (hexane extract—IC50 14.94 μg/mL; chloroform—IC50 15.74 μg/mL; ethyl acetate—IC50 16.18 μg/mL; ethanol—IC50 19.43 μg/mL). In comparison with ethyl acetate extract from *Gracilaria verrucosa,* ethyl acetate extract from *Hypnea flagelliformis* showed anti-cancer activity with a much higher IC50 value, of about 138.321 μg/mL. 

Ashwini et al. [86] also used chloroform and ethanol extracts from marine seaweed *Gracilaria corticata* to illustrate its anticancer potency against HeLa cell lines. The cytotoxic activity was evaluated at 24 h, 48 h and 72 h. Chloroform and ethanol extracts showed a significant anti-cancer activity (IC50 341.82 µg/mL and 244.7 µg/mL, respectively, after 48 h), and the morphology of the treated cells was different in comparison with the control cells. 

According to a recent report of Gomes et al. [87], methanolic extracts from 13 species of seaweeds collected from the Northeast of Brazil showed different levels of cytotoxicity against SiHa and HeLa cells. The most potent anti-cancer activities were exhibited by *Dictyota cilliolata* and *Dictyota menstrualis*. Fluorescence microscopy showed significant biochemical and morphological changes of the cells exposed to *Dictyota cilliolata* and *Dictyota menstrualis,* such as chromatin condensation, phosphatidylserine externalization and loss of cell viability. Furthermore, *Dictyota cilliolata* methanolic extracts induced cell cycle arrest in phase S. 

*Turbinaria conoides* can be found on the east coast of the Gulf of Thailand, and it is known for its potency in suppressing tumor cells. Fresh samples of this seaweed were used to treat HeLa cells. The extracts of *Turbinaria conoides* inhibited HeLa cells in a dose-dependent manner (IC50 20.92 ± 3.15 μg/mL). The investigators also observed that the most exposed cells possessed apoptotic nuclei in comparison to unexposed cells [88]. 

Vaseghi and collab. [89] investigated the cytotoxic activity of *Sargassum angustifolium*, a seaweed collected from the southwestern coastline of the Persian Gulf. They used a methanolic extract of this seaweed in order to evaluate its effects on HeLa cervical cells after 72 h from the exposure. The results indicated that the survival rate of HeLa cells was decreased by increasing the concentration of extracts from 150 μg/mL to 900 μg/mL. This report suggested the high potential of *Sargassum angustifolium* as a potential anti-cancer drug. 

Kim et al. [90] were the first to investigate the biological activity of Antarctic red seaweed *Iridaea cordata* on cervical cancer. They extracted sulfated polysaccharides from *Iridaea cordata* and showed their cytotoxic activity against HeLa cells. In comparison with other species of seaweeds, *Iridaea cordata* exhibited a weaker anticancer activity against HeLa, Pc-3 and HT-29 tumoral cells. 

Silva Costa et al. [91] performed a study in order to evaluate the antioxidant and anticancer activity of five different heteroflucans extracted from *Sargassum filipendula.* Their results clearly indicated a strong antioxidant capacity and a dose-dependent cytotoxic effect against HeLa cells (IC50 15.69 μg/mL). Moreover, Arsianti et al. [92] tested the cytotoxic activity of *Euchema cotonii* extracts using HeLa cells. Chloroform, n-hexane, ethanol, and ethyl acetate extracts of *Euchema cottonii* exhibited a high anti-tumoral effect against cervical HeLa cells with values of IC50 of 4.82 μg/mL, 5.73 μg/mL, 7.54 μg/mL, and 4.34 μg/mL, respectively.

Fucoxanthin is a water-soluble dietary fiber extracted from alginic edible seaweeds that exhibited a more significant cytotoxic effect against HeLa cells in comparison with other candidates. Ye et al. [93] observed that after the treatment with fucoxanthin for 24 h, the proteins of apoptotic markers significantly changed in HeLa cells. Furthermore, fucoxanthin considerably decreased NF-kB activation and suppressed tumoral growth in vivo, proving that it may be used as an anti-cervical cancer therapeutic agent in the future. Another study conducted in order to investigate whether and how fucoxanthin exerts cytotoxic effects on HeLa cells was conducted by Hou et al. [94] in the year 2013. HeLa cells were treated with fucoxathin in doses ranging between 10 and 80 μmol/L, for 48 h. The results indicated that fucoxanthin induced dose-dependent cytotoxicity (IC50—55.1 ± 7.6 μmol/L) and G0/G1 arrest. Furthermore, fucoxanthin increased the expression of PTEN and decreased the concentrations of Akt and p53 protein in a dose-dependent manner. The development of cervical cancer was associated with the PI3K/Akt signaling pathway. In many types of cancer, various alterations in the upstream regulators of Akt (such as phosphatidylinositol 3-kinase (PI3K)) were described. For this reason, it is possible that PI3K or Akt inhibitors may be used as potent therapeutic agents in the fight against cervical cancer. 

Tumor necrosis factor-related apoptosis-inducing ligand (TRAIL) is a cytokine able to induce apoptosis in many types of tumoral cells. Jin et al. [95] separately analyzed the effects of TRAIL and fucoxanthin on HeLa, CaSki and SiHa cells. Their results indicated that the highest effectiveness in terms of apoptosis belonged to the combination of TRAIL and fucoxanthin. These molecules produced a synergistic effect on apoptosis in human cervical cancer cells. Furthermore, fucoxanthin was able to suppress the signaling PI3K/Akt and NF-κB pathways-mediated cell apoptosis, while TRAIL activated these pathways. 

Shaik and collab. [96] used methanol, hexane and buthanol as solvents to dissolve crude extracts of *Ulva fasciata, Sargassum wightii* and *Gracillaria corticata,* in order to analyze their cytotoxic potential against three different cell lines, including cervical cancer cells. Their findings showed that a butanolic extract of *Gracillaria corticata* exhibited a stronger anticancer activity than hexane and methanolic extracts of the same seaweed. Moreover, the butanolic extract of *Gracillaria corticata* was more efficient as a cytotoxic agent against HeLa cells than hexane and methanolic extracts of *Sargassum wightii* and *Ulva fasciata*. 

Another study that pointed out the cytotoxic effect of *Sargassum wightii* on HeLa cells was conducted by Suganya et al. [97]. They synthesized silver nanoparticles using a methanolic extract of *Sargassum wightii*, and incubated them with human cervical cancer cell lines for 24 or 48 h. The investigators reported that the biologically synthesized silver nanoparticles exhibited dose-dependent cytotoxicity against HeLa cells. Additionally, Asik and collab. [98] synthesized hexagonal-shaped zinc oxide nanoparticles, using aqueous extracts of *Gracilaria edulis*, and incubated them with cervical carcinoma cells (SiHa cells). Zinc oxide nanoparticles exhibited cytotoxic effects against SiHa cells (IC50 35 ± 0.03 μg/mL) in a dose-dependent manner. Furthermore, these nanoparticles induced ROS-mediated, mitochondrial-dependent apoptosis and necrosis in SiHa cells. 

Table 2 summarizes the results of the studies presented above.

## 6. Conclusions

Seaweeds are known for their richness in bioactive molecules, proteins, fatty acids, minerals and vitamins, and their daily consumption may solve the problems of various nutritive principles deficiencies in human nutrition. In this systematic review, we illustrated the most relevant studies from the recent literature regarding the effectiveness of edible algae as anti-cervical cancer and anti-HPV agents. Human dietary studies do show, on average, trends in favor of seaweed ingestion in the protection from cancer development, but the effect is small. 

The advantages of seaweeds consumption, such as wide acceptability, low costs and low cytotoxicity in normal cells, make them great candidates for the development of novel antiviral and cytotoxic drugs. In the literature, there is a lack of studies regarding the effects of seaweeds compounds on premalignant cervical lesions, where their effects could be more relevant. We hope that our study will pave the way to further clinical trials on this topic. Instead, several studies on cervical cancer cell lines are described, in which high doses of polysaccharides extracted from seaweeds have been shown to be effective as cytotoxic agents.

According to the scientific evidence, high concentrations of polysaccharides are necessary for the prevention of HPV infection—much higher that what could be achieved with dietary supplementation. In our opinion, dietary supplementation with edible seaweeds is not sufficient for the prevention of HPV infection. However, HPV vaccines, which are protective against multiple strains of the virus, are already available against HPV infection. Vaccination with Gardasil^®^ and Cervarix^®^ targets preadolescent girls and young women. A recent analysis of FUTURE I and FUTURE II clinical trials has shown that including older women in the vaccination program might reduce re-infection and might prevent HPV persistent infection in naïve women [100]. Under these conditions, we consider that the association between vaccination and dietary supplementation with seaweeds could be a stronger tool against HPV infection in comparison with dietary supplementation alone. 

Therefore, considering all the beneficial results previously reported, we consider seaweeds a topic of major interest for the biomedical community in the field of cervical cancer and HPV infection. Additionally, future in vivo and in vitro studies targeting premalignant cervical lesions need to be conducted in order to establish the doses and timing of the administration of seaweeds compounds for the prevention of these lesions and for the prevention of carcinogenesis development.

## Figures and Tables

**Figure 1 ijms-22-00629-f001:**
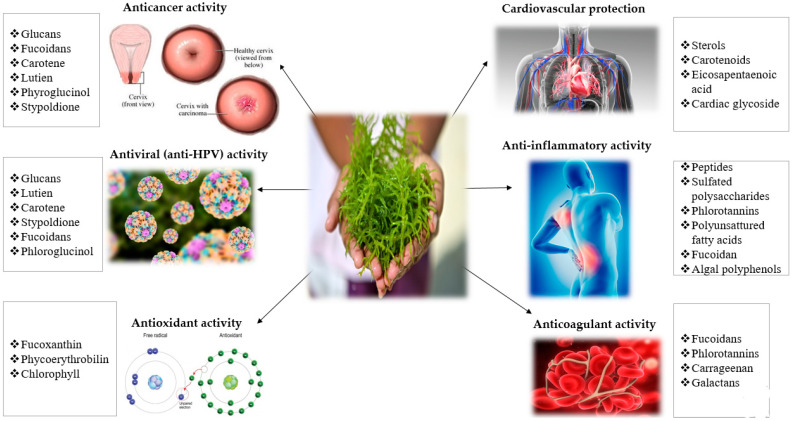
Health-promoting effects of seaweeds and their bioactive compounds [11].

**Figure 2 ijms-22-00629-f002:**
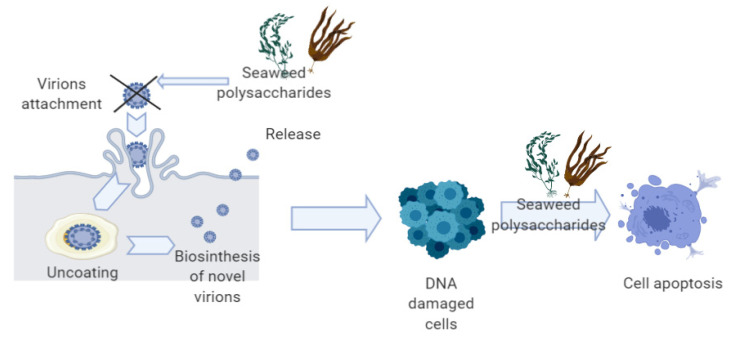
Anti-HPV and pro-apoptotic mechanisms of seaweeds polysaccharides.

**Table 1 ijms-22-00629-t001:** Bioactive compounds from seaweeds.

Author, Reference	Seaweed	Bioactive Compound
Matsubara, 2004 [35]	*Codium cylindricum*	Sulfated galactan
Jurd, 1995 [36]	*Codium fragile*	Arabinogalactans
Vasanthi, 2004 [37]	*Corallina pilulifera*	Ethanolic extract
Yang, 2009 [38]	*Ecklonia cava*	Phlorotannins
Shibata, 2002 [39]	*Eisenia bicyclis*	Phloroglucinol
Yan, 1999 [40]	*Hijikia Fusiformis*	Fucoxanthin
Park, 2011 [41]	*Lobophora variegate*	Fucans
Chevolot, 1999 [42]	*Phaeophyceae*	Fucoidans
Yabuta, 2010 [43]	*Porphyra spp.*	Phycoerythrobilin
Kaza-owska, 2010 [44]	*Porphyria dentate*	Cathecol
Subba Rao, 1997 [45]	*Rhodophyceae*	Galactans
Shibata, 2002 [39]	*Saccharina japonica*	Fucoidans
Bae, 2011 [46]	*Sargassum thunbergii*	Phlorotannins
Vasanthi, 2004 [37]	*Schizymenia dubyi*	Sulfated glucuronogalactan
Mayer, 2000 [47]	*Taonamaria atomaria*	Stypoldione
Yan, 1999 [40]	*Undaria pinnatifida*	Fucoxanthin

**Table 2 ijms-22-00629-t002:** Studies of anti-HPV and anti-cervical cancer effects of seaweeds extracts.

STUDIES ON ANTI-HPV EFFECT					
Author, Reference	Health-Promoting Effect	Seaweeds	Bioactive Compounds	Methods	Results
Rodriguez et al., 2014 [81]	Anti-HPV		Carrageenan	Carrageenan 3% was applied as intravaginal gel in a mouse model, infected with HPV16, HPV18 and HPV45 pseudoviruses. In vitro, PC-515 gel containing carrageenan was tested on a murine model.	Carrageenan formulations were effective as anti-HPV agents both in vitro and in vivo. Physicochemical properties of carrageenan lubricants influence the protection against HPV infection in vivo.
Santos et al., 2019 [83]	Anti-HPV	*Porphyra umbilicalis*		The diet of 44 mice infected with HPV16 was supplemented with 10% seaweed. After the study period, the skin of the animal models was examined in order to classify HPV16-induced lesions.	*Porphyra umbilicalis* dietary intake significantly reduced the incidence of dysplastic lesions in HPV16 infected mice.
Wang et al., 2020 [82]	Anti-HPV	Brown seaweeds	Polymannuroguluronate sulfate (PMGS)	The effects of PMGS were tested in mice and in vitro, using HeLa cells.	PMGS inhibited skin infection with HPV in mice. PMGS allowed the downregulation of E6 and E7 proteins of HPV. PMGS targeted L1 protein from the viral capsid and blocked the HPV infection.
**STUDIES ON ANTI-CERVICAL CANCER EFFECTS**					
Saengkhae et al., 2010 [88]	Antiproliferative effects Cytotoxic effects	*Turbinaria conoides*		HeLa cells were treated with fresh samples of *T. conoides*.	*T. conoides* exhibited cytotoxic effects against HeLa cells (IC50 20.92 ± 3.15 μg/mL) in a dose-dependent manner.
Costa et al., 2011 [91]	Antiproliferative effects Antioxidant effects	*Sargassum filipendula*	Fucans	Antiproliferative effects of *S. filipendula* were tested in vitro, on HeLa, HepG2 and PC-3 cells.	The strongest antiproliferative effect was exhibited against HeLa cells. *S. filipendula* also exerted antioxidant effects in vitro.
Hou et al., 2013 [94]	Pro-apoptotic effects		Fucoxanthin	HeLa cells were used to evaluate the cytotoxicity of fucoxanthin (doses ranged between 10 and 80 μmol/L), for 48 h.	Fucoxanthin induced G0/G1 arrest in a dose-dependent manner, and also increased the expression of LC3 II (autophagosome marker). Fucoxanthn inhibited the phosphorylation of Akt and increased PTEN expression in tumoral cervical cells.
Ye et al., 2014 [93]	Pro-apoptotic effects		Fucoxanthin	HeLa cells were treated with fucoxanthin for 24 h.	Fucoxanthin induced apoptosis in tumoral cells. The phosphorylation of Akt significantly decreased depending on the dose of fucoxanthin. Fucoxanthin interfered with the mitochondrial signal transduction pathway.
Shaik et al., 2014 [96]	Cytotoxic effects	*Sargassum wightii Ulva fasciata Gracillaria corticata*		Methanolic, butanolic and hexanoic extracts of selected seaweeds were used on HeLa cells and their cytotoxic effects were registered.	Butanolic extracts of *G. corticata* showed the most potent cytotoxic activity.
Kim et al., 2016 [90]	Cytotoxic effects	*Iridaea cordata*	Sulfated polysaccharides	HeLa, HT-29 and PC-3 cells were treated with polysaccharides extracted from *Iridaea cordata* seaweed.	*Iriaea Cordata* polysaccharides exerted a weak antitumor activity against HeLa, Pc-3 and HT-29 tumoral cells.
Ashwini et al., 2016 [86]	Cytotoxic effects	*Gracilaria corticata*	Chloroform and ethanol extracts	The anticancer activity of chloroform and ethanol extracts from seaweeds on HeLa cells was observed after 24, 48 and 72 h.	IC50 for chloroform extracts 341.82 µg/mL. IC50 for ethanol extracts 244.7 µg/mL.
Arsianti et al., 2018 [92]	Cytotoxic effects	*Eucheuma cottonii*	Ethanol, chloroform, and ethyl acetate extracts	Cytotoxic activity of ethanol, chloroform, and ethylacetate extracts from *Eucheuma cottonii* were tested on HeLa cells.	IC50 for ethanol extract—7.54 μg/mL. IC50 for ethylacetate extract—4.34 μg/mL. IC50 for chloroform extract—4.82 μg /mL.
Micheylla et al., 2018 [85]	Cytotoxic effects	*Gracilaria verrucosa*	Hexane, chloroform, ethyl acetate, ethanol extracts	Seaweed extracts were diluted into 8 concentrations and their anticancer activity was investigated in vitro, using HeLa cells.	The most potent cytotoxic activity against HeLa cells was exhibited by hexane extract. IC50 for hexane extract 14.94 μg/mL. IC50 for chloroform extract 15.74 μg/mL. IC50 for ethyl acetate 16.18 μg/mL. IC50 for ethanol extract 19.43 μg/mL.
Vaseghi et al., 2018 [89]	Cytotoxic effects	*Sargassum angustifolium*	Methanol-ethyl acetate extracts	HeLa and MCF-7 cells were treated with methanol-ethyl acetate extracts from *Sargassum angustifolium* (150, 450, and 900 μg/mL).	Seaweed extracts exhibited cytotoxic effects against HeLa and MCF-7 cells in a dose-dependent manner.
Jin et al., 2018 [95]	Pro-apoptotic effects		Fucoxanthin	HeLa cells were treated with fucoxanthin or TRAIL (Tumor necrosis factor-related apoptosis-inducing ligand)	The combination of TRAIL and fucoxanthin induced apoptosis in HeLa cells Fucoxanthin inhibited PI3K/Akt and NF-κB pathways for apoptosis. TRAIL upregulated PI3K/Akt and NF-κB pathways in HeLa cells.
Raubbin et al., 2018 [99]	Cytotoxic effects	*Hypnea flagelliformis*		The effects of different concentrations of ethyl acetate extract from *H. flageliformis* on HeLa cells were observed after 48 h of incubation.	IC50 for ethyl acetate extract was 138.321 μg/mL.
Suganya et al., 2019 [97]	Pro-apoptotic effects	*Sargassum wightii*		Silver nanoparticles containing seaweed *S. wightii* extracts were incubated for 24 or 48 h with HeLa cells. Cytotoxic activity of silver nanoparticles with different concentrations was investigated.	Silver nanoparticles showed a significant decrease in cell viability after 24 h. The complete loss of cell viability was observed after 48 h. Silver nanoparticles showed a 50% inhibition of HeLA viability at 47.48 μg/mL in 24 h.
Asik et al., 2019 [98]	Cytotoxic and pro-apoptotic effects	*Gracilaria edulis*		Zinc nanoparticles were synthetized using an aqueous extract from *Gracilaria edulis* seaweed. Their anticancer effects were investigated on SiHa cells.	IC50 for zinc nanoparticles—35 ± 0.03 μg/mL. Zinc nanoparticles induced apoptosis and necrosis in SiHa cells in a dose-dependent manner.

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
