# Peer review of "Are Bioactive Molecules from Seaweeds a Novel and Challenging Option for the Prevention of HPV Infection and Cervical Cancer Therapy?—A Review"

_ijms, 2021, doi:10.3390/ijms22020629_

Round 1

Reviewer 1 Report

The submitted manuscript focuses on the anticancer and antiviral activities exerted by a wide range of polysaccharides obtained from marine algae. These activities are described mostly with regard to cervical carcinoma.

Although this Review is original and its contents could be of interest to the biomedical community, the data summarized herein are reported and discussed in a chaotic, repetitive and incomplete way.  This is particularly evident with regard to the effects that polysaccharides have on the molecules involved in carcinogenesis. Furthermore, little importance has been attached to the fact that the vast majority of the studies reported in this Review were performed in vitro, in cell lines derived from highly progressed cervical carcinoma, where antitumor effects were observed at high doses of the polysaccharides. In this regard, little attention has been paid to preneoplastic lesions of the uterine cervix, where the effects of the polysaccharides mentioned here could be more relevant.

The best part of the Review is its Table 2. In my opinion, the manuscript should be completely rearranged, using Table 2 as a template to describe the information in a more thorough and orderly way.

Author Response

Dear Reviewer,

We thank you for your cooperation and we appreciate you taking the time to analyze our work. Following your comments and suggestions, we made some revisions to our paper, in order to have a clearer presentation of our results, as follows:

  • We shortened the abstract in order to make it more precise
  • Extensive editing of English language and style was realized
  • Section 2- we completed this section and reorganized it
  • Section 3- this chapter has been divided into two sub-sections: anti-HPV mechanisms and anti-cervical cancer mechanisms of action of seaweeds. We revised the presented information and completed it, in a more comprehensive way
  • Section 5 was also revised and rearranged, taking table 2 as template. We presented the studies according to the two major inclusion criteria: studies based on the effectiveness of seaweed in the prevention of HPV and studies on their effectiveness in the treatment of cervical cancer.
  • Table 2 was separated into 2 sections
  • We also revised the conclusions trying to give a more personal note to this chapter. We also mentioned an aspect that was not discussed yet in the literature: the effects of the association between the HPV vaccine and the consumption of seaweeds, against HPV infection.
  • Regarding the prenoplastic lesions of the uterine cervix, in the literature there is only one study conducted on this topic, and we presented it in the Results section.

Reviewer 2 Report

The review “Are Bioactive Molecules from Seaweeds a Novel and Challenging Option for the Prevention of HPV Infection and Cervical Cancer Therapy? – A Review” is a well-designed methodical paper ad presents all the published data compiled in a lucid manner. It would certainly be of interest to the readers. However, I have a few questions:

  1. As also pointed by the authors, sea weeds form a major art of traditional diets of many Asian cultures. Is there any scientific evidence showing less incidence of cervical cancers in these populations? If there is, the authors should include a paragraph about that and also include some statistics if possible about incidence and prognosis of cervical cancers in these populations.
  2. Do the authors believe that dietary supplementation will be sufficient preventive measure against cervical cancers/HPV infections? Because the concentrations of bioactive compounds used in referenced studies is much higher that what could be achieved with dietary supplementation.
  3. Since HPV vaccines are already available against HPV which are protective against multiple strains of the virus, what is the significance of dietary bioactive compounds as a preventive measure? Please include a paragraph in the conclusion.
  4. Minor corrections: The abstract should be shortened and made more precise. There are also few spelling mistakes and language errors which should be revised.

Author Response

Dear Reviewer,

We thank you for your cooperation and we appreciate you taking the time to analyze our work. Following your comments and suggestions, we made some revisions to our paper, in order to have a clearer presentation of our results, as follows:

Point 1:

As also pointed by the authors, sea weeds form a major art of traditional diets of many Asian cultures. Is there any scientific evidence showing less incidence of cervical cancers in these populations? If there is, the authors should include a paragraph about that and also include some statistics if possible about incidence and prognosis of cervical cancers in these populations.

Response 1:

There are no scientific evidence regarding the incidence of cervical cancer in these populations, and we included a paragraph about this subject in the Introduction

Point 2:

Do the authors believe that dietary supplementation will be sufficient preventive measure against cervical cancers/HPV infections? Because the concentrations of bioactive compounds used in referenced studies is much higher that what could be achieved with dietary supplementation.

Point 3:

Since HPV vaccines are already available against HPV which are protective against multiple strains of the virus, what is the significance of dietary bioactive compounds as a preventive measure? Please include a paragraph in the conclusion.

Response 2+3

We do not believe that dietary supplementation with marine algae is enough for the prevention of HPV infection. In our opinion, there are many factors involved in the prevention, and dietary supplementation is a valuable tool for the prevention but it is not sufficient. We discussed this aspect in the Conclusions section

Point 4:

Minor corrections: The abstract should be shortened and made more precise. There are also few spelling mistakes and language errors which should be revised.

Response 4:

We shortened the abstract in order to make it more precise and extensive editing of English language and style was realized

Reviewer 3 Report

In this review, the authors collected information regarding the potential application of seaweeds active components for the treatment or prevention of HPV infection, which is a leading cause of cervical cancer. The abstract nicely summarizes the work but since the introduction, the flow of information resulted a bit confused. For instance, in line 50 the authors said “the degrees of risk for cervical cancer secondary to the infection with various types of HPV have not been properly assessed yet” but in line 62 they stated “Almost 70% of vaginal, 43% of vulvar and 100% of cervical tumors are associated with HPV infection” thus being contradictory, apparently. In line 91 anticoagulant is repeated. Going on to section 2, seaweeds and their active components are described. In line 105 “its” and “has” have to be replaced with “their” and “have” since seaweeds is plural. In lines 107 and 110 a similar list of active components is reported, so they could be merged. CO2 is not subscript. In line 115 “mainly” should be “principal”. In line 149 heparin sulfate should be heparan?

Beside these little mistakes, sections 3 and 5 are not clearly organized. They sound more like a list of compounds with some details on their activity rather than a critical dissertation on the subject, as it should be instead. Section 5 in particular should provide a more detailed discussion about the reported literature and not only a brief list of findings. I do understand that the final table can help in following the story, but it looks like the whole manuscript lacks of discussion and perspectives. Moreover, there are already a few paper dealing with seaweeds active polysaccharides (e.g. Carbohydrate Research, 2017, 453–454, 1-9; IJPCBS 2016, 6(3), 271-279)

However, I do believe that this subject would be of interest for the journal audience. Therefore, I would suggest to rewrite the main sections following a more critical overview and furnishing more detailed information, in order to increase the originality of the paper and submit it again for further revision.

Author Response

Dear Reviewer,

We thank you for your cooperation and we appreciate you taking the time to analyze our work. Following your comments and suggestions, we made some revisions to our paper, in order to have a clearer presentation of our results, as follows:

Point 1:

The abstract nicely summarizes the work but since the introduction, the flow of information resulted a bit confused. For instance, in line 50 the authors said “the degrees of risk for cervical cancer secondary to the infection with various types of HPV have not been properly assessed yet” but in line 62 they stated “Almost 70% of vaginal, 43% of vulvar and 100% of cervical tumors are associated with HPV infection” thus being contradictory, apparently. In line 91 anticoagulant is repeated. Going on to section 2, seaweeds and their active components are described. In line 105 “its” and “has” have to be replaced with “their” and “have” since seaweeds is plural. In lines 107 and 110 a similar list of active components is reported, so they could be merged. CO2 is not subscript. In line 115 “mainly” should be “principal”. In line 149 heparin sulfate should be heparan?

Response 1:

  • The contradictory information was removed
  • Extensive editing of English language and style was realized
  • CO2 was subscript

Point 2:

Beside these little mistakes, sections 3 and 5 are not clearly organized. They sound more like a list of compounds with some details on their activity rather than a critical dissertation on the subject, as it should be instead. Section 5 in particular should provide a more detailed discussion about the reported literature and not only a brief list of findings. I do understand that the final table can help in following the story, but it looks like the whole manuscript lacks of discussion and perspectives. Moreover, there are already a few paper dealing with seaweeds active polysaccharides (e.g. Carbohydrate Research, 2017, 453–454, 1-9; IJPCBS 2016, 6(3), 271-279)

However, I do believe that this subject would be of interest for the journal audience. Therefore, I would suggest to rewrite the main sections following a more critical overview and furnishing more detailed information, in order to increase the originality of the paper and submit it again for further revision.

Response 2:

- We shortened the abstract in order to make it more precise

- Section 2- we completed this section and re-organized it

- Section 3- this chapter has been divided into two sub-sections: anti-HPV mechanisms and anti-cervical cancer mechanisms of action of seaweeds. We revised the presented information and completed it, in a more comprehensive way

- Section 5 was also revised and rearranged, taking table 2 as template. We presented the studies according to the two major inclusion criteria: studies based on the effectiveness of seaweed in the prevention of HPV and studies on their effectiveness in the treatment of cervical cancer.

- Table 2 was separated into 2 sections

- We also revised the conclusions trying to give a more personal note to this chapter. We also mentioned an aspect that not yet discussed in the literature: the effects of the association between the HPV vaccine and the consumption of seaweeds, against HPV infection.

Round 2

Reviewer 1 Report

The revised version of the present review describes scientific findings in a quite orderly and logically sequential way. Sufficient attention has now been paid to the effects that some of the constituents of marine algae have on the molecular mechanisms of carcinogenesis. The anti-HPV activities of the above mentioned compounds are clearly reported. However, there is still some repetition between what is described in chapter 3 and in chapter 5. Such repetitions should be eliminated before the manuscript is published.

Author Response

According to the reviewers’ suggestions, we made the following modifications of our manuscript:

  • We eliminated the repetitions between the sections
  • We replaced the word "sulphate(d)" with "sulfate(d)" throughout the text
  • Regarding the words “heparin/heparan”, they have two different means: heparin is a glycosaminoglycan extracted from seaweeds, while heparan is a molecule from the cell surface, which is able to bind HPV. These two words are used throughout the text depending on their mean.

Reviewer 3 Report

In the second version of their manuscript, the authors significantly improved the quality and the organisation of the work. They added several references and included a more detailed discussion of the subject. Also the idea of dividing into subsections created a clearer flow of information. Repetitions have been avoided  and conclusions are more exhaustive. 

I would just recommend to check one last time "sulfate/sulphate" which is written sometimes with the "f" and sometimes with the "ph", along with "heparin/heparan" which are sometimes exchanged. Please, be careful while accepting the corrections in the file as some format settings might be lost. Same comment for the literature.

The manuscript is now acceptable for publication after the above-mentioned little corrections 

Author Response

(The authors gave the same response as above.)
